# Regulation of Let-7a-5p and miR-199a-5p Expression by Akt1 Modulates Prostate Cancer Epithelial-to-Mesenchymal Transition via the Transforming Growth Factor-β Pathway

**DOI:** 10.3390/cancers14071625

**Published:** 2022-03-23

**Authors:** Abdulrahman Alwhaibi, Varun Parvathagiri, Arti Verma, Sandeep Artham, Mir S. Adil, Payaningal R. Somanath

**Affiliations:** 1Clinical and Experimental Therapeutics, University of Georgia, Augusta, GA 30912, USA; aalwhaibi@ksu.edu.sa (A.A.); vparvathagiri@augusta.edu (V.P.); averma2@augusta.edu (A.V.); sandeep.artham@duke.edu (S.A.); madil@augusta.edu (M.S.A.); 2Charlie Norwood VA Medical Center, Augusta, GA 30912, USA; 3Clinical Pharmacy Department, College of Pharmacy at King Saud University, Riyadh 11451, Saudi Arabia; 4Vascular Biology Center, Augusta University, Augusta, GA 30912, USA; 5Georgia Cancer Center, Augusta University, Augusta, GA 30912, USA

**Keywords:** Akt1, miR-199a-5p, let-7a-5p, epithelial-to-mesenchymal transition, prostate cancer

## Abstract

**Simple Summary:**

The molecular mechanisms regulating the switch from the growth of tumor cells to invasive phenotype for metastasis is largely unknown. Molecules such as Akt1 and TGFβ have been demonstrated to play reciprocal roles in the early and advanced stages of cancers, and epithelial-to-mesenchymal transition has been identified as a common link in the process. Advancing our knowledge on the direct association between these two pathways and how their effects are reconciled in the advanced stages of cancers such as prostate cancer will have therapeutic benefits. Identifying the role of microRNAs in the process will also benefit the scientific community.

**Abstract:**

Akt1 suppression in advanced cancers has been indicated to promote metastasis. Our understanding of how Akt1 orchestrates this is incomplete. Using the NanoString^®^-based miRNA and mRNA profiling of PC3 and DU145 cells, and subsequent data analysis using the DIANA-mirPath, dbEMT, nCounter, and Ingenuity^®^ databases, we identified the miRNAs and associated genes responsible for Akt1-mediated prostate cancer (PCa) epithelial-to-mesenchymal transition (EMT). Akt1 loss in PC3 and DU145 cells primarily induced changes in the miRNAs and mRNAs regulating EMT genes. These include increased miR-199a-5p and decreased let-7a-5p expression associated with increased TGFβ-R1 expression. Treatment with locked nucleic acid (LNA) miR-199a-5p inhibitor and/or let-7a-5p mimic induced expression changes in EMT genes correlating to their anticipated effects on PC3 and DU145 cell motility, invasion, and TGFβ-R1 expression. A correlation between increased miR-199a-5p and TGFβ-R1 expression with reduced let-7a-5p was also observed in high Gleason score PCa patients in the cBioportal database analysis. Collectively, our studies show the effect of Akt1 suppression in advanced PCa on EMT modulating miRNA and mRNA expression changes and highlight the potential benefits of miR-199a-5p and let-7a-5p in therapy and/or early screening of mPCa.

## 1. Introduction

The American Cancer Society estimates 268,490 new cases and 34,500 deaths from prostate cancer (PCa) in 2022 [1]. PCa has the highest incidence of cancer among men, and the mortality in PCa, the second leading cause of cancer-related deaths among men in the United States [2], is primarily due to its metastasis [3,4] and lack of reliable screening methods [5]. The major known risk factors for PCa are age, ethnicity, obesity, and family history [6], in addition to the genetic background, ethnicity, and family history [7,8]. PCa starts as pre-malignant lesions of high-grade prostatic intraepithelial neoplasia (High-Grade PIN), which then advances to the tumor, nodal, and metastatic (PNM stages) PCa [9]. Mechanistically, how metastatic PCa (mPCa) emerges from the primary tumors is a current gap in our knowledge base, in turn being a major roadblock in its therapeutic development and the primary reason for the high mortality. Digital rectal examination and serum prostate-specific antigen, the two gold-standard screening methods used for PCa detection and follow-up, have very low sensitivity and high non-specificity [10]. Therefore, there is an urgent need for more accurate diagnostic and predictive tools as well as improved therapies for mPCa. 

Transforming growth factor-β (TGFβ) is a well-known epithelial growth suppressor from which solid tumors arise [10,11]. Although a tumor suppressor early on, TGFβ switches into a promoter of cancer cell invasion and metastasis in the advanced stages [12,13]. Whereas TGFβ induces apoptosis in some PCa cells [14], the advanced stage PCa cells adapt to TGFβ stimuli and differentiate into mesenchymal cells to evade the growth-suppressive effects of TGFβ [11,15] by a process known as epithelial-to-mesenchymal transition (EMT) [16]. Our laboratory has demonstrated both the growth-suppressive [14] and EMT-inducing effects of TGFβ in PCa [17]. During EMT, epithelial cells shun the epithelial markers, such as E-cadherin and keratins, and acquire expression of novel proteins such as N-cadherin, vimentin, smooth muscle cell actin-ɑ, and EDA-Fibronectin, and transcription factors such as Snail and Zeb1/2, etc. [18]. EMT, the process of early-stage cancer cell differentiation to a malignant phenotype, is a pre-requisite for cancer cell invasion [19] and dissemination [20,21]. 

Interestingly, we noticed a paradoxical correlation between the TGFβ and the Akt pathways in the differential regulation of early and late-stage PCa. Being a growth suppressor, Akt inhibition in PCa [13,22] and endothelial cells [23] by TGFβ did not surprise us. However, the finding that suppression of Akt1 is a crucial step in the acquisition of PCa EMT and promotion of metastasis was remarkable [24]. Similar effects were observed in Akt1-deficient endothelial cells promoting endothelial-to-mesenchymal transition [25,26] and promoting PCa metastasis [27]. Furthermore, human PCa tissues with a high Gleason score showed reduced phosphorylated Akt1 expression/activity and an enhanced TGFβ level compared to lower Gleason score tissues [28]. Interestingly, pharmacological or genetic suppression of Akt1 led to the modulation of genes belonging to the TGFβ superfamily [24,29,30], indicating the existence of an Akt1-TGFβ signaling axis in PCa metastasis. A paradoxical effect of Akt1 suppression on metastasis has also been reported in other types of cancers [31] but with limited molecular insights.

MicroRNAs (miRNAs or miRs) are small non-coding RNAs that post-transcriptionally regulate gene expression by binding to the 3′ untranslated regions (3′ UTR) of the target mRNAs [32]. Whereas a small portion of miRNAs are encoded by their genes, most of them come from large coding and non-coding transcripts known as pre-miRs that are processed within the nuclear by DROSHA and DGCR8 [33,34,35]. These pre-miRs are translocated to the cytosol by Exportin-5, where another ribonuclease, Dicer, cleaves the pre-miRs into ~20 bp-long, mature miRNAs [36]. Finally, the RNAS-induced silencing complex (RISC) mature the miRNA duplex and guide it to the target mRNA for gene silencing [36]. A review of the literature detailing the mechanisms of miRNA biosynthesis and the potential role of Akt-regulated miRNAs in cancer was recently published by our group [35]. 

Several recent studies have implicated the importance of miRNAs in the regulation of TGFβ signaling [37,38], EMT, and cancer metastasis [39], including PCa [30,40]. Studies also support the use of miRNAs as prognostic or predictive biomarkers of PCa [41]. However, having discovered over 2000 miRNAs in humans that regulate one-third of the total gene pool in the genome [42], our understanding of the miRNAs regulating cancer metastasis is still in the primitive stage. Akt1 has been demonstrated to regulate miR-200a-5p in TGFβ-induced EMT in human breast cancer cells [43]. Apart from this, the Akt substrate FoxO is a master regulator of miRNAs in cancer cells [35,44]. Hence, it is important to profile the Akt1-regulated miRNAs in mPCa and study their involvement in the modulation of TGFβ pathway.

In the current study, on a NanoString^®^ platform, we profiled Akt1-regulated miRNAs in human PC3 and DU145 mPCa cells and identified miR-199a-5p and let-7a-5p as key Akt1-regulated miRNAs in the activation of the TGFβ pathway and promotion of EMT. Bioinformatics analysis and data mining from the cBioportal database confirmed increased miR-199a-5p and decreased let-7a-5p expression in high Gleason score compared to low Gleason score PCa tissues associated with increased TGFβ type-I receptor (TGFβ-RI) expression. Treatment with miR-199a-5p inhibitor and/or let-7a-5p mimic diminished TGFβ-RI expression and EMT marker expression in PC3 and DU145 cells, in turn inhibiting their migration and invasion. Our study identified novel Akt1-TGFβ crosstalk regulated by miRNAs and supports the use of miR-199a-5p and let-7a-5p for early detection and treatment of mPCa. 

## 2. Materials and Methods

### 2.1. Cell Culture, shRNA Gene Silencing, Antibodies, and Other Reagents

Androgen-independent human PCa (PC3 and DU145) cells were purchased from ATCC (Manassas, VA, USA), grown, and maintained in DMEM high-glucose medium (Hyclone, Logan, UT, USA) with 10% FBS (Atlanta Biologicals, Flowery Branch, GA, USA), 100 U/mL penicillin, and 100 mg/mL streptomycin in a humidified incubator at 37 °C and 5% CO_2_. Gene silencing was performed using SMARTvector 2.0 Lentivirus ShControl (non-targeting) and ShAkt1 (5′-3′: ACGCTTAACCTTTCCGCTG) from Dharmacon (Lafayette, CO), followed by selection with 0.6 ng/mL puromycin (Sigma, St. Louis, MO, USA). Anti-TGFβ-R1 antibodies (Cat#31013) were purchased from Abcam (Cambridge, MA, USA). Antibodies for Snail (Cat#3879), E-cadherin (Cat#3195), N-cadherin (Cat#4061), and cleaved caspase-3 (Cat#9664) were purchased from Cell Signaling (Danvers, MA, USA). We chose PC3 and DU145 cells for this study because of our previous experience with regard to the downregulation of the Akt pathway [24,27,45], the loss of AR expression in Akt1 deficient cells [46], and because LNCaP cells did not respond to TGFβ stimulation to induce EMT [45].

### 2.2. NanoString-Based miRNA and mRNA Profiling

ShControl and ShAkt1 cells were used for the miRNA and mRNA profiling. Briefly, cells were plated in 6-well plates until 60–70% confluence, washed 2 times with 1X PBS, and lysed for RNA isolation using RNeasy Plus Mini Kit (Qiagen, Hilden, Germany, Cat#73414) according to the manufacturer’s instructions. The RNA purity and concentration were determined using a NanoDrop (Thermo Scientific, Waltham, MA USA) and the quality of miRNA was analyzed using an Agilent 2100 Bioanalyzer. An nCounter gene fusion panel (NanoString Technologies, Inc., seattle, WA, USA) was used to determine the copy numbers of miRNA and mRNA targets in each sample according to the manufacturer’s instructions. Changes in the miRNA and mRNA targets were normalized to a set of housekeeping genes in the designed panels. After normalization, positive ratio values indicated higher gene expression in ShAkt1 compared to Shctrl cells, while negative values indicated higher genes expression in Shctrl compared to the ShAkt1 group. All ratios were presented with *p* values. Only significantly changed miRNA and mRNA were considered for further analysis. For volcano plots, the fold change values were converted into their respective log2-fold change values. These log2-fold change values were plotted against the -log10 *p* values with the fold change cutoff of 1.2 and *p*-value cutoff of 0.05 and further analyzed using R version 4.1.1.

### 2.3. In Vivo miRNA Isolation and Quantitative RT-PCR

miRNAs were extracted from tumor nodules isolated from the lungs of athymic nude mice that were injected with either Shctrl or ShAkt1 DU145 cells and sacrificed after 16 days from the injection. miRNAs extraction was carried out using an miRNeasy Mini Kit (Cat# 1038703) and RNeasy MiniElute Cleanup Kit (Cat#74204) purchased from Qiagen and according to the manufacturer’s protocols. A Nanodrop analyzer (Thermo Scientific, Inc., Waltham, MA USA) was used to test RNA sample purity and concentration. cDNA synthesis was performed using the miScript II RT Kit (Cat#218161, Qiagen, Hilden, Germany). miScript SYBR^®^ Green PCR Kit (Cat#218075, Qiagen) was used for qRT-PCR. The primers for *hsa*-miR-199a-5p is (5′-3′: CCCAGUGUUCAGACUACCUGUUC) and the primer for *hsa*-let-7a-5p (5′-3′: UGAGGUAGUAGGUUGUAUAGUU), and was purchased from Qiagen. RNU6 (RNA, U6 small nuclear 2) and SNORD61 (small nucleolar RNA, C/D box) were used as the normalization reference genes to analyze the expression of miRs.

### 2.4. miRNA Transfection

Locked Nucleic Acid (LNA) technology developed, highly stable and efficient human miR-199a-5p inhibitor (5′-3′: CCCAGUGUUCAGACUACCUGUUC; Cat#YI04101096), let-7a-5p mimic (5′-3′: UGAGGUAGUAGGUUGUAUAGUU; Cat#YM00470408), and their negative controls (Cat#YI00199006 and Cat#YM00479902, respectively) were purchased from Qiagen. miRNA transfection was carried out using lipofectamine 2000 (Invitrogen, Waltham, MA, USA) in Opti-MEM (Thermo Scientific, Waltham, MA, USA). Cells were incubated in 6-well plates until 60–70% confluent. A final concentration of 25 nM miR-199a-5p inhibitor and 10 nM let-7a-5p mimic were transfected into the target cells and further incubated for a total of 72 h.

### 2.5. Western Blotting

Western blotting was performed as published previously [27]. Briefly, the whole-cell lysates were prepared using 1X RIPA lysis buffer (Millipore, Temecula, CA, USA) supplemented with protease and phosphatase inhibitor tablets (Roche Applied Science, Indianapolis, IN, USA). Protein concentration was measured by the DC protein assay (Bio-Rad Laboratories, Hercules, CA, USA), and approximately 30 ug to 40 µg of cell lysates mixed with 2X Laemmle’s buffer were used. Densitometry was performed using NIH ImageJ software. The original WB can be found at Appendix A original images.

### 2.6. mirPath v.3 and dbEMT Analysis

KEGG (Kyoto Encyclopedia of Genes and Genomes) analysis on DIANA-mirPath v.3 databases (http://diana.imis.athena-innovation.gr/, accessed on 20 May 2019) was performed to predict all targets genes or signaling pathways regulated by miRNAs. dbEMT (EMT gene database) analysis was performed specifically to look into the role of known or potential genes/targets, which are regulated by each or combination of the profiled microRNAs, obtained from the KEGG and miR-Path on EMT and cancer progression/metastasis (http://dbemt.bioinfo-minzhao.org, accessed on 14 May 2020).

### 2.7. Migration Assay

The migration assay was performed as previously published [47]. Cells were grown on 6-well plates until reaching 70–80% confluence. Scratch was made in the cell monolayer using a 1 mL pipette tip followed by a one-time wash with 1X PBS. Cells were incubated after miRNA transfection in serum-free DMEM high-glucose medium for 24 h then replaced with full medium thereafter. Images of scratches were taken immediately after scratching (0 h) and 48 h after transfection. The rate of migration was measured as a percentage of scratch filling using the equation ((1 − T48/T0) × 100), where T48 is the area at the endpoint (48 h) and T0 is the area measured immediately after making the scratches.

### 2.8. Matrigel^®^ Invasion Assay

Matrigel invasion assay was performed as previously reported from our laboratory [17]. Briefly, 24-well Transwell^®^ permeable plate supports with 8.0 μm polycarbonate membrane and Matrigel^®^ were purchased from Corning Life Sciences (Tewksbury, MA, USA). A concentration of 5 mg/mL of Matrigel was used for coating supports. Briefly, 24 h after the miRNA transfection process, explained previously in the migration assay, the medium was aspirated, and cells were washed once with 1X PBS, detached using sterile 20 mM EDTA in PBS, and washed once with PBS. Cells were re-suspended in serum-free medium. Using a Countess automated cell counter (Invitrogen), 100,000 cells were seeded onto the Matrigel^®^ in the upper chamber of the Transwell plates filled with 100 µL of serum-free medium. Cells that invaded the Matrigel and reached the bottom layers of the supports after 48 h incubation were fixed using 3.7% paraformaldehyde and then stained with 0.5% crystal violet solution. Three bright-field images of each insert were taken using an inverted microscope and three blinded reviewers counted the invaded cells. The average number of invaded cells from every three images was calculated and considered for the analysis.

### 2.9. Ingenuity^®^ Pathway Analysis

Ingenuity Pathway Analysis (IPA, Qiagen Bioinformatics) is a software that transforms a list of molecules into a set of relevant networks associated with pathology based on extensive records maintained in the Ingenuity Pathways Knowledge Base [30]. Highly interconnected networks are predicted to represent significant biological functions. IPA was used to map the significantly changed/shared miRNAs associated with Akt1 inhibition in both cell lines to genome-wide association study (GWAS)-implicated cancer genes along with molecular canonical/non-canonical pathways and biological functions observed with various cancer. All genes that were directly affected by the upregulated or downregulated miRNAs were subjected for further analysis.

### 2.10. cBioportal Genomic Data Analysis of the MSKCC Study

Data on the expression (log2 normalized) of miR-199a-5p and let-7a-5p as well as the corresponding clinical information published by the MSKCC team [48], the only dataset that contained miRNA information, were obtained from cBioPortal (https://www.cbioportal.org/, 10 April 2019). miR-199a-5p and let-7a-5p were analyzed in the dataset. To investigate the correlation of miRNAs expression with Gleason scores, patients with a Gleason score 6 or 7 (*n* = 87) were grouped and compared to those with a Gleason score 8 or 9 (*n* = 17). The student t-test was used for the analysis of differential expression of miR-199a-5p and let-7a-5p between the two cohorts.

### 2.11. Statistical Analysis

All the data are presented as the mean ± standard deviation (SD) to determine significant differences between treatments and control values. We used one-way ANOVA for groups of 3 or more and Student’s two-tailed t-test for studies including 2 independent groups. Statistical analysis was performed using GraphPad Prism version 6.01 software and the results are considered significant when *p*-value < 0.05.

## 3. Results 

### 3.1. Akt1 Inhibition Changes the microRNA Expression Profile in PC3 and DU145 Cells

Expression profiling of 827 miRNAs in Akt1-silenced PC3 and DU145 human PCa cells was carried out using Nanostring to identify the actual changes in miRNA copy numbers with Akt1 gene ablation (Figure 1 and Figure 2A, Appendix A). Out of the mapped miRNAs, 222 and 125 miRNAs changed their expression significantly (*p* < 0.05) in DU145 and PC3 cells, respectively, from which a total of 61 miRNAs showed matched expression. 

Bioinformatics analysis using Ingenuity^®^ was performed on the matched upregulated and downregulated miRNAs to identify the mRNAs that are potentially involved in the regulation of cancer cell EMT as well as processes involved in cancer metastasis (Appendix A, Figure 2B,C and Figure 3). Out of the 35 downregulated miRNAs, five miRNAs were linked to the genes regulating the TGFβ pathway. Among these, only let-7a-5p showed a negative correlation with the expression of TGFβ1 and TGFβ-R1 as well as the cancer cell migration and metastasis. Three out of the 26 upregulated miRNAs were linked to the TGFβ pathway. From these, only miR-199a-5p showed a positive correlation with TGFβ-R1, TGFβ-R2, proangiogenic gene expression, and invasive carcinoma. Overall, the two aggressive Akt1-silenced PCa cell lines, miR-199a-5p upregulation and let-7a-5p downregulation, were found to be potentially involved in the regulation of the TGFβ pathway (Figure 1). 

A separate NanoString mRNA profiling (Figure 4 and Figure 5) and volcano plot analysis revealed significant modulation of genes regulating the TGFβ pathway and EMT (Figure 6A,B). Further analysis of the NanoString-profiled mRNA changes compared to the predicted miRNA-regulated EMT genes from the dbEMT database revealed common hits that belong to the TGFβ pathway that is involved in the regulation of EMT (Figure 6C).

### 3.2. Human PCa Tissues with a High Gleason Score Express Higher miR-199a-5p and Lower Let-7a-5p Compared to the Low Gleason Score Tissues

To confirm that let-7a-5p and miR-199a-5p expressions were modulated in Akt1-deficient PCa cells, we collected control and Akt1-deficient DU145 tumor colonies from the mouse lungs previously administered with these cells via the tail vein. Analysis of these colonies revealed decreased let-7a-5p and increased miR-199a-5p expressions with Akt1 gene deletion (Figure 7A). 

To validate the clinical relevance of our findings, miR-199a-5p and let-7a-5p expression was detected in human PCa samples from cBioportal analysis (*n* = 9–11), which showed, although not significant, an elevation of miR-199a-5p accompanied by a reduction of let-7a-5p in a high Gleason score (GS 6–7) compared to low Gleason score (GS 8–10) samples (Figure 7B,C). Furthermore, miR-199a-5p and let-7a-5p expression were determined in the MSKCC study dataset of human PCa specimens with two different cohorts of high and low Gleason scores. Clinicopathological characteristics of 113 patients with miRNAs are summarized in Figure 7D,E. A total of 104 patients were classified into two groups based on their low (*n* = 87) and high (*n* = 17) Gleason scores. The expression of miR-199a-5p was significantly elevated and let-7a-5p significantly reduced in the high compared to the low Gleason score group (Figure 7F,G). These results suggested that the elevation of miR-199a-5p and inhibition of let-7a-5p expression could be linked to higher Gleason score PCa and, potentially, tumor cell EMT and metastasis. Further analysis of mRNAs based on the median miR-199a-5p and let-7a-5p expression confirmed the significant positive correlation between miR-199a-5p and TGFβ-R1 expression (Figure 8A,B) with a trend toward the elevation of TGFβ1 expression with higher miR-199a-5p levels (Figure 8C). Surprisingly, however, increased let-7a-5p expression was associated with a significant increase in TGFβ-R1 with a trend toward inhibition of TGFβ1 expression (Figure 8D–F) with no changes in Akt1 expression in any conditions (Appendix A).

### 3.3. Transfections with miR-199a-5p Inhibitor and/or let-7a-5p Mimic Impairs Migration and Invasion of PC3 and DU145 Cell Migration

The overexpression of let-7a-5p mimic or miR-199a-5p inhibitor in PC3 and DU145 cells resulted in impaired migration in a 48-h monolayer scratch assay (Figure 9A–D). Based on our TargetScan 7.2 software analysis, we identified that let-7a-5p has a direct target site on TGFβ-R1 mRNA (Appendix A). Transfection of DU145 cells with miR-199a-5p inhibitor (25 nM) and/or let-7a-5p mimic (10 nM) was performed to determine their effect on TGFβ-R1 expression. Western blotting results showed that overexpression of let-7a-5p mimic or miR-199a-5p inhibitor in DU145 cells results in reduced expression of TGFβRI (Figure 10A) and increased expression of epithelial E-cadherin (Figure 10B), compared to non-target miRNA control. Overexpression of the let-7a-5p mimic or miR-199a-5p inhibitor in DU145 cells also resulted in impaired invasion in a Matrigel^®^-based assay system (Figure 10C,D), with no significant effect on cell viability assessed by an MTT assay (Figure 10E). Similar effects were also observed in PC3 cells (Appendix A). There were no significant differences between single transfections with the let-7a-5p mimic or miR-199a-5p inhibitor in PCa cells compared to combined transfections on TGFβ-R1 expression, invasion, and migration.

## 4. Discussion

Although unexpected, a flurry of recent reports from various laboratories has demonstrated that Akt1 suppression in advanced cancers worsens the condition by augmenting the process of EMT, in turn promoting metastasis [31]. Interestingly, despite different molecular mechanisms reported by various laboratories for this unforeseen role of Akt1 in advanced cancers, it does get reconciled at one single point, which is EMT [31]. Elevated expression of EMT transcription factors, such as ZEB and Snail/Slug, EMT markets, such as N-cadherin, and smooth muscle cell actin-α were corroborated with the activation of the TGFβ super family members such as TGFβ1 and Nodal [24,29], potentially regulated by the microRNAs [30,35]. 

miRNAs are one of the first set of molecules proposed to promote EMT in the absence of Akt1 expression in a breast cancer study [43]. Unfortunately, further research on the Akt1-miRNA axis in any other cancers was not performed since its first report in 2009. We recently demonstrated the role of the Akt-miRNA axis in the potential regulation of EMT in a TRAMP mouse model of PCa [30]. Whereas miR200a was the predominant Akt1-regulated miRNA identified in breast cancer cells, this, however, was unchanged in the Akt-suppressed TRAMP prostates suggesting that the Akt1-regulated miRNAs in different cancer types may vary based on their origin and/or specific mutations that they hold. Furthermore, there are differences between human and mouse miRNAs in their nomenclature as well as their function and molecular regulation. Hence, it is important to profile the Akt1-regulated miRNAs in advanced cancers and study their molecular and functional implications in advanced cancers for translational purposes.

Our findings on Akt1 suppression modulating miRNA and mRNA expressions to promote EMT in human PC3 and DU145 cells were supported by the bioinformatics analyses of cellular data and patient databases. Ingenuity pathway analysis predicted increased miR-199a-5p and/or reduced let-7a-5p because of Akt1 suppression in PC3 and DU145 cells responsible for the promotion of EMT, both of which reciprocally regulated the TGFβ signaling pathway. The TargetScan analysis identified a direct target of let-7a-5p on TGFβ-R1 mRNA, suggesting that a reduced let-7a-5p would promote EMT by activating the TGFβ signaling pathway. Interestingly, we did not detect a direct target for miR-199a-5p on any of the TGFβ pathway molecules, suggesting that the effect of miR-199a-5p on PCa cell EMT could be due to the secondary effects. The genomic analysis of PCa patient samples using the data available from the cBioportal database indicated increased miR-199a-5p and reduced let-7a-5p in a high Gleason score compared to the low Gleason score PCa tissues. Transfections with either miR-199a-5p inhibitor or let-7a-5p mimic resulted in impaired motility and invasion of PCa cells. Collectively, our study revealed that miR-199a-5p and let-7a-5p regulated by Akt1 in the advanced PCa modulate the TGFβ signaling pathway, in turn regulating EMT.

Literature indicates the importance of miRNAs in PCa [49,50]. Previous reports from other laboratories on the role of let-7a-5p in cancer metastasis are highly in agreement with our findings in this study. Let-7a-5p, in combination with another long-coding RNA, has been shown to develop doxorubicin resistance in DU145 PCa cells [51]. Downregulation of let-7a-5p expression has been reported to predict lymph node metastasis and its prognosis in colorectal cancer [52]. Concomitant downregulation of let-7a-5p and let-7f-5p miRNAs have been reported in the plasma and stool samples collected from early-stage colorectal carcinoma [49,53]. In the current study, we show for the first time that the under-expression of let-7a-5p leads to EMT in the advanced PCa is regulated by Akt1. Furthermore, our study has identified TGFβ-R1 as the primary target of let-7a-5p in PCa cells, where overexpression with let-7a-5p mimic was observed to inhibit TGFβ-R1 expression, PCa cell migration, and invasion. TGFβ-R1 as the direct target of let-7a-5p was also confirmed by TargetScan software analysis. Overall, this finding is a very important step forward in our knowledge on how Akt1 downregulation in the advanced PCa suppresses let-7a-5p expression, in turn promoting PCa cell EMT via increased TGFβ-R1 expression.

Unlike let-7a-5p, the literature on the potential role of miR-199a-5p in cancer is conflicting and its potential role in the regulation of EMT in cancer is not very clear. A recent study has shown reduced expression of miR-199a-5p in PCa tissues compared to the normal prostate and BPH tissues [54]. This is highly in agreement with our findings that mir-199a-5p expression correlates with Akt1 activity. It has been demonstrated that Akt1 activity is higher in primary prostate tumors compared to normal prostate tissues [24,55]. Hence, a high Akt1 activity correlating to the reduced miR-199a-5p expression in primary PCa tissue is not surprising. This might also be a cancer cell adaptation to keep the TGFβ pathway in check, which otherwise functions as a tumor suppressor in early cancers [11]. However, when the Akt1 activity declines as the cancers progress to the advanced stages, an increase in the miR-199a-5p expression is also expected, which is evident in Akt1-deficient PC3 and DU145 cells. cBioportal analysis and the RT-PCR analysis of patient samples show a positive correlation between miR-199a-5p and TGFβRI expression. The TGFβ-R1 expression was also reduced when PC3 or DU145 cells were transfected with miR-199a-5p inhibitor. However, a direct target for miR-199a-5p on TGFβ-R1 was not identified in TargetScan software analysis, suggesting that the effect of the miR-199a-5p inhibitor on TGFβ-R1 expression could be secondary. Nevertheless, our study shows for the first time that Akt1 suppression in the advanced PCa increases miR-199a-5p expression, in turn promoting TGFβ-R1 expression in PC3 or DU145 cells in the regulation of cell motility and invasion.

## 5. Conclusions

In conclusion, our data showed that miR-199a-5p elevation and let-7a-5p inhibition resulting from Akt1 suppression in the advanced PCa activates the TGFβ pathway through increased TGFβ-R1 expression, thus promoting PCa cell EMT (Appendix A). One of the limitations in the study is that the experiments are focused on characterizing the molecular mechanisms in cellular models; hence, conformational studies will be necessary for further validation. In addition, while our study reveals the association between Akt1 and miRNAs in the regulation of the TGFβ pathway, EMT, and PCa cell motility and invasion, the Akt1-regulated miRNAs and mRNAs modulating other signaling pathways and cellular processes will need further investigation. Nevertheless, our results signify the translational importance of the identified miRNAs for mPCa. The elevated miR-199a-5p and reduced let-7a-5p expression in the high Gleason score human PCa tissues and the ability of the miR-199a-5p inhibitor or let-7a-5p mimic to inhibit PCa cell migration and invasion demonstrate the potential of miR-199a-5p and let-7a-5p in therapy and early screening of mPCa.

## Figures and Tables

**Figure 1 cancers-14-01625-f001:**
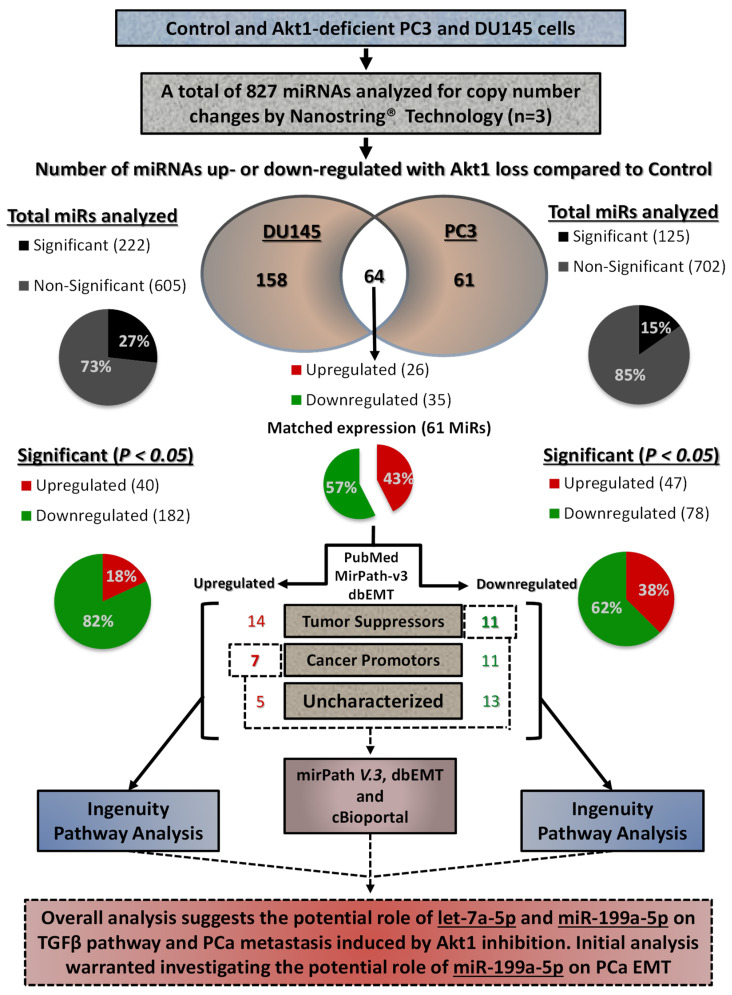
Flow chart describing the profiling of Akt1-regulated miRNAs in human PC3 and DU145 PCa cells. Among the 827 miRNAs analyzed using the NanoString technology, 222 miRNAs in DU145 cells and 125 miRNAs in PC3 cells were found to be modulated by Akt1 gene silencing. Out of the 61 matched miRNAs from DU145 and PC3 cells with Akt1 loss (26 up- and 35 downregulated), there were 7 upregulated cancer promoters and 11 downregulated cancer suppressors, which were subjected for bioinformatics analysis using the Ingenuity^®^, mirPath V.3, TargetScan, dbEMT, and cBioportal databases to identify miR-199a-5p and let-7a-5p as the key Akt1-regulated miRNAs modulating PCa cell EMT.

**Figure 2 cancers-14-01625-f002:**
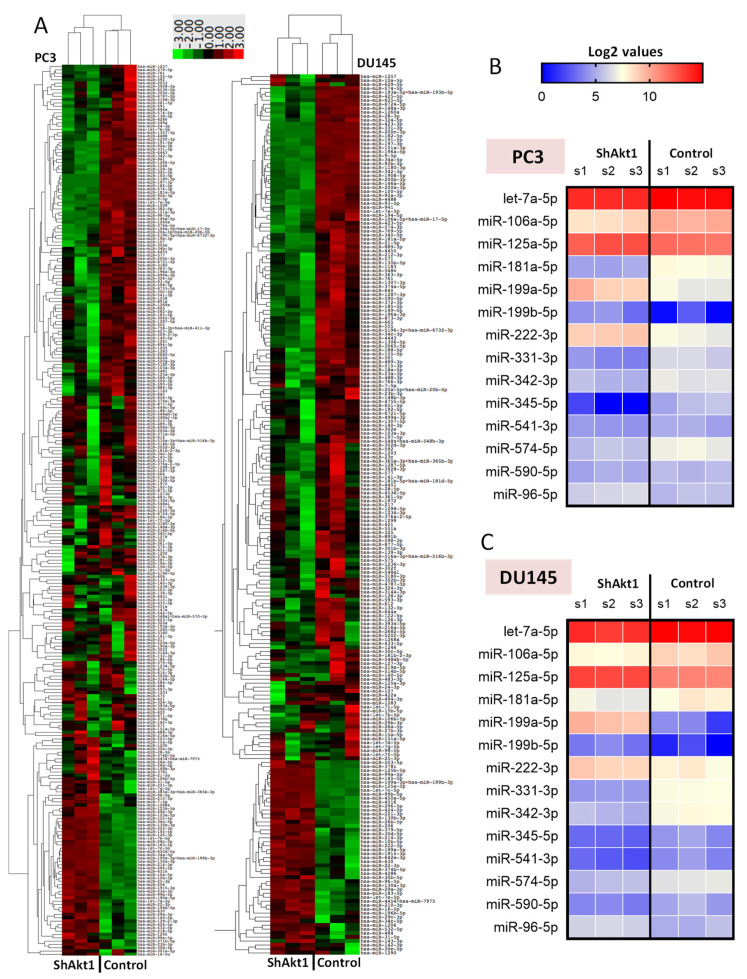
Heatmaps showing miRNA expression changes in DU145 and PC3 cells with Akt1 gene silencing. (**A**) All 827 miRNAs and (**B**,**C**) modulated miRNAs regulating EMT in PC3 and DU145 cells, respectively. Only changes in the miRNAs that are statistically significant (*p* < 0.05) are shown (*n* = 3).

**Figure 3 cancers-14-01625-f003:**
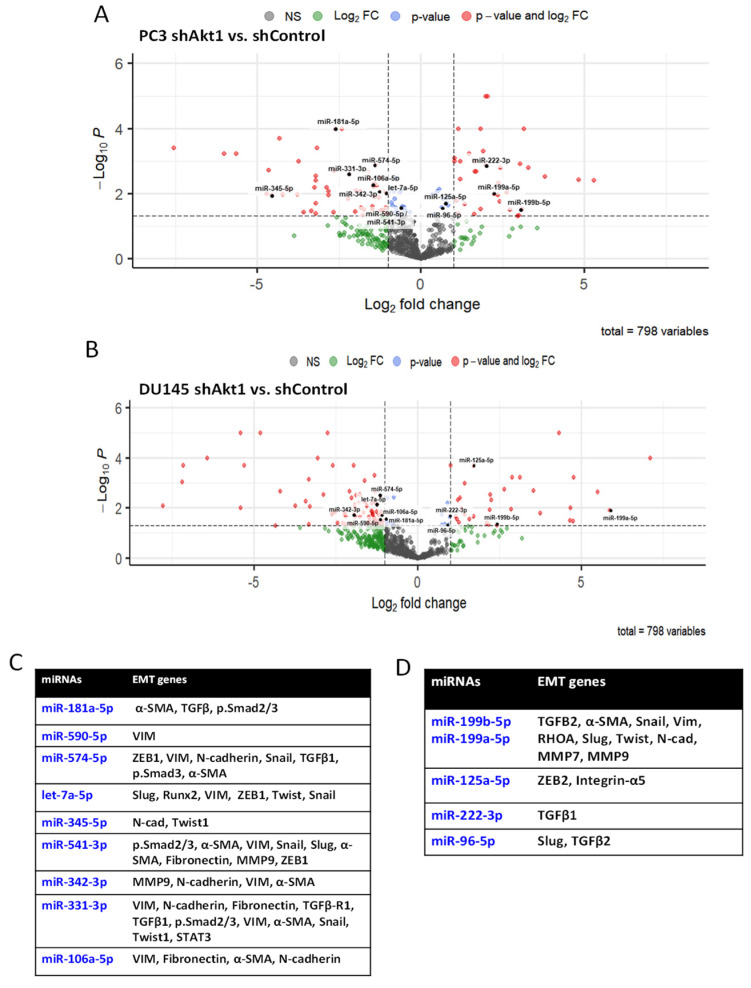
KEGG and Gene Ontology (DIANA-mirPath) and dbEMT database analyses indicate modulations of EMT and metastasis-regulated genes by miRNAs in Akt1-deficient PC3 and DU145 cells. (**A**,**B**) Volcano plot showing highly modulated, EMT regulating miRNAs in Akt1-deficient PC3 and DU145 cells compared to Akt1 intact controls. (**C**,**D**) Chart showing predicted and known targets of the identified miRNAs from NanoString analysis of Akt1-deficient PC3 and DU145 cells indicating their predominant involvement in the regulation of EMT and metastasis. Only changes in the miRNA that are statistically significant (*p* < 0.05) are shown (*n* = 3).

**Figure 4 cancers-14-01625-f004:**
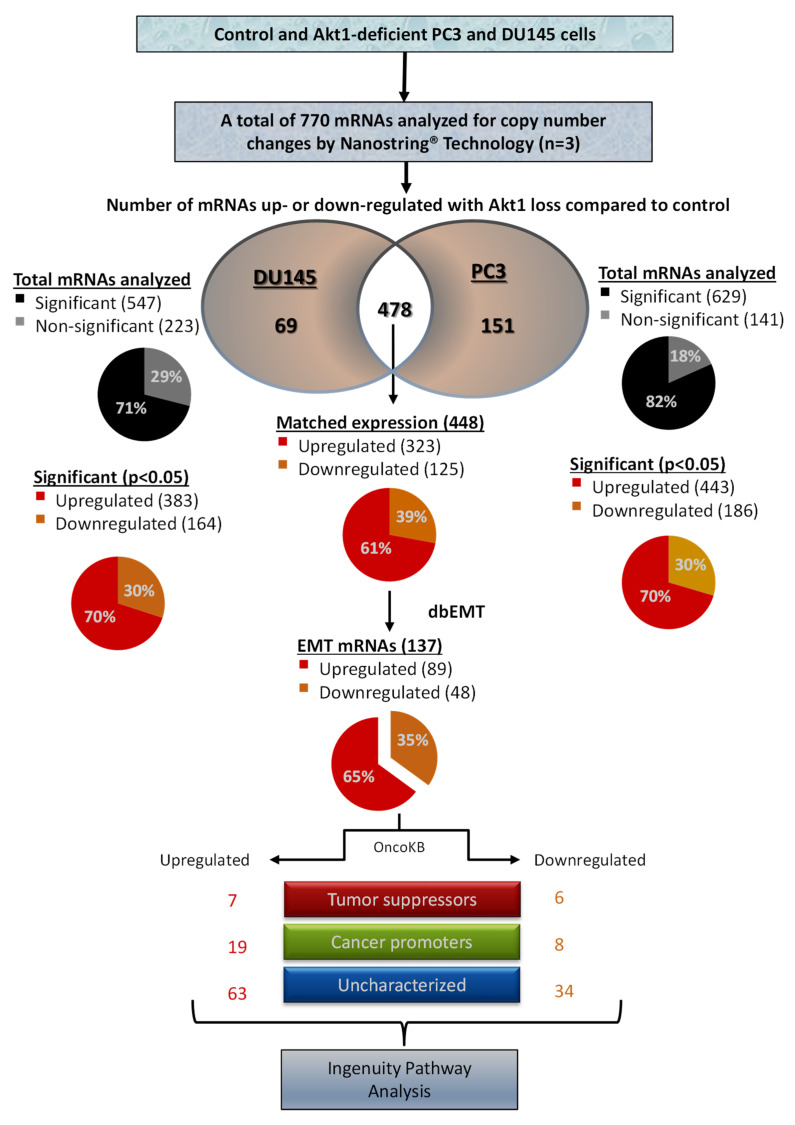
Flow chart describing the profiling of Akt1-regulated mRNAs in human PC3 and DU145 PCa cells. Among the 770 mRNAs analyzed using the NanoString^®^ technology, 547 mRNAs in DU145 cells and 629 mRNAs in PC3 cells were found to be modulated by Akt1 gene silencing. Out of the 448 matched mRNAs from DU145 and PC3 cells with Akt1 loss (323 up- and 125 downregulated), there were 19 upregulated cancer promoters and 6 downregulated cancer suppressors, which were subjected for bioinformatics analysis using the Ingenuity, dbEMT, and cBioportal databases to identify the key Akt1-regulated miRNAs modulating PCa cell EMT.

**Figure 5 cancers-14-01625-f005:**
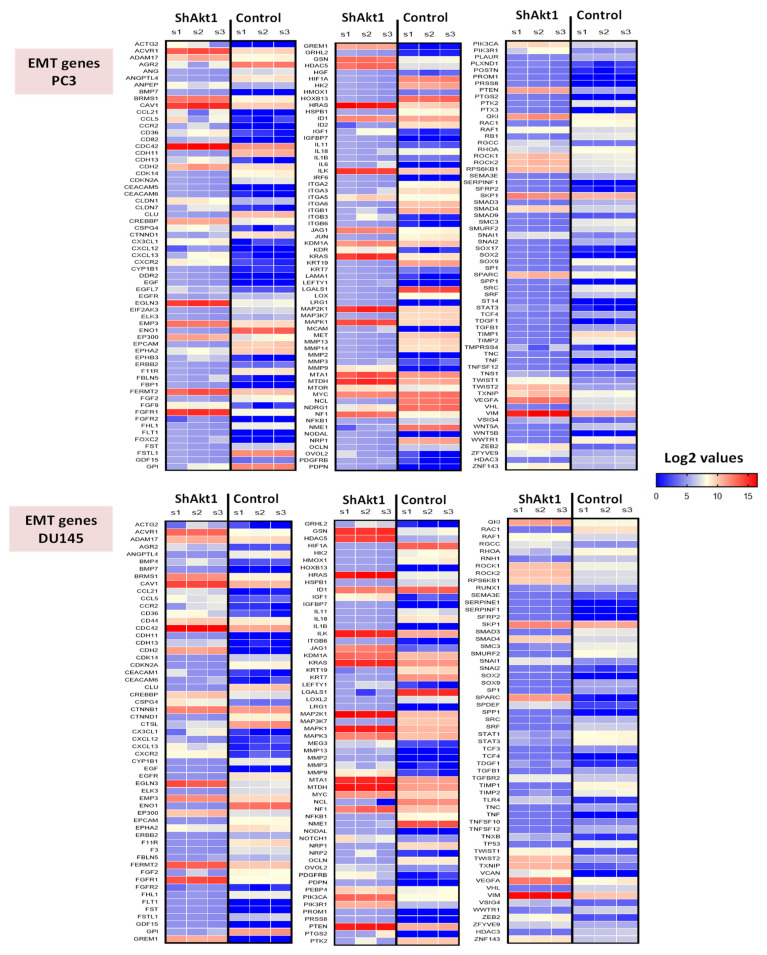
Heatmaps of selective mRNAs that are modulated in PC3 and DU145 PCa cells upon shRNA mediated Akt1 knockdown compared to shControl. Only changes in the mRNA that are statistically significant (*p* < 0.05) are shown (*n* = 3).

**Figure 6 cancers-14-01625-f006:**
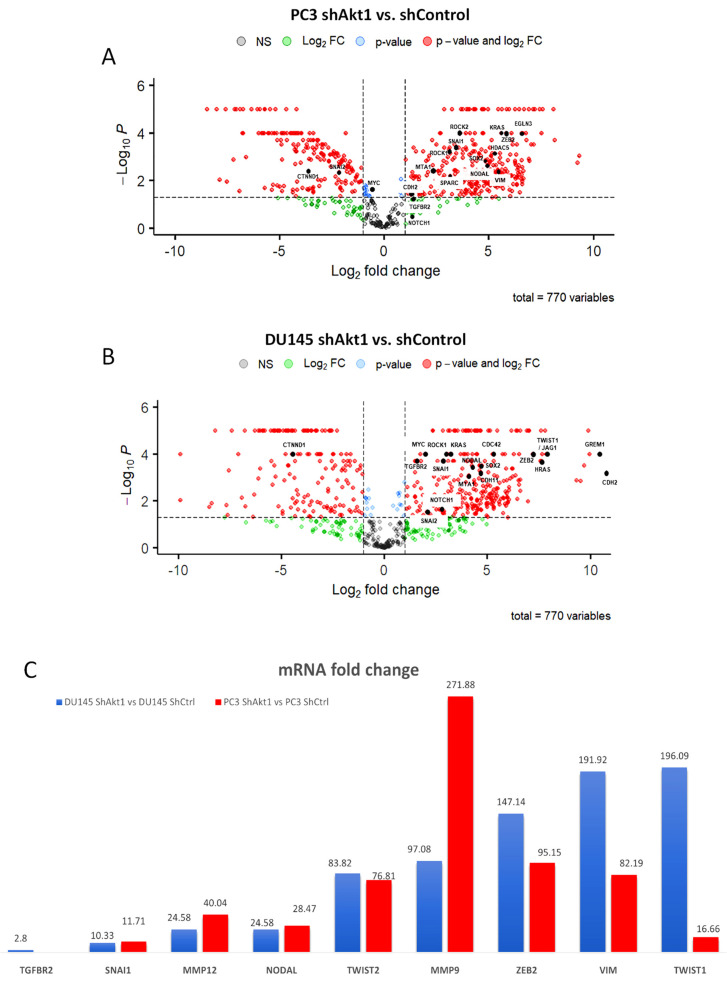
Nanostring analysis, indicating modulations in EMT and metastasis regulating mRNAs in Akt1-deficient PC3 and DU145 cells. (**A**,**B**) Volcano plot showing highly modulated, EMT regulating mRNAs in Akt1-deficient PC3 and DU145 cells compared to Akt1 intact controls. (**C**) Bar graph showing mRNAs identified to be modulated by Akt1 deficiency in PC3 and DU145 cells compared to the predicted and known targets from dbEMT analysis. Only changes in the mRNA that are statistically significant (*p* < 0.05) are shown (*n* = 3).

**Figure 7 cancers-14-01625-f007:**
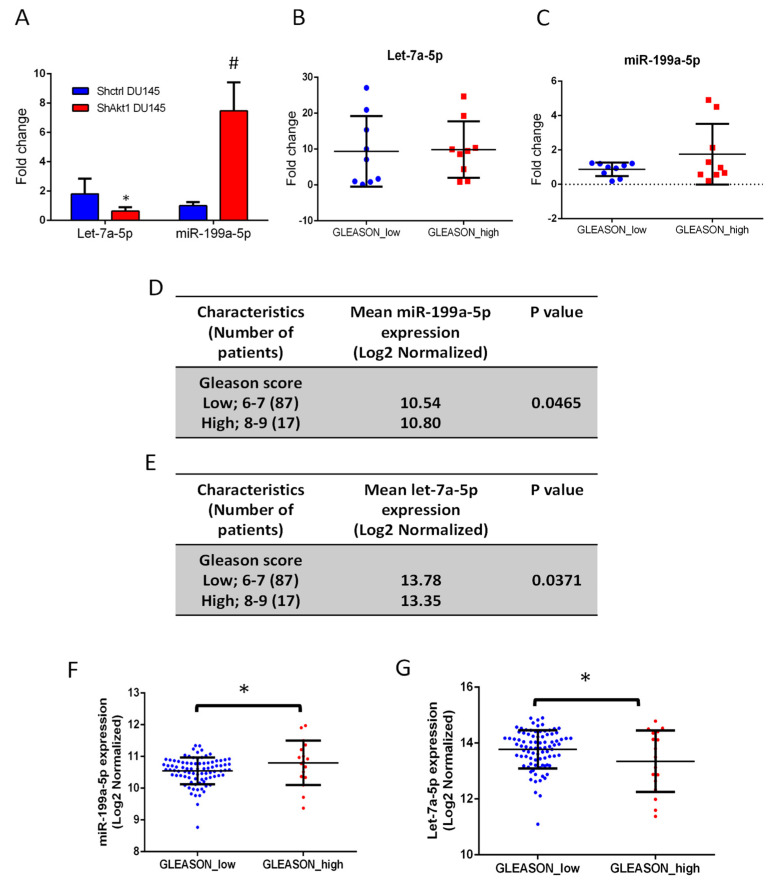
miR-199a-5p and let-7a-5p expression changes correlate with PCa Gleason score. (**A**) Quantitative real-time PCR data for miR-199a-5p and let-7a-5p in the DU145 colony collected from the mouse lungs previously administered (via the tail vein) with ShControl and ShAkt1 DU145 cells showing increased miR-199a-5p and decreased let-7a-5p expression (*n* = 5; fold-change compared to SNORD61 expression). (**B**,**C**) Quantitative real-time PCR data for miR-199a-5p and let-7a-5p in high Gleason score PCa tissues compared to low Gleason score (*n* = 9–11), respectively. (**D**,**E**) Data showing the mean normalized log2 miR-199a-5p and let-7a-5p expression changes, respectively, in the high and low Gleason score cohorts. In total, 104 out of 113 patients were included in the analysis (seven were excluded with unknown Gleason Scores). Comparison between the two cohorts showing that miR-199a-5p expression is significantly upregulated in the higher (8–9) compared to lower Gleason score (6–7) samples (N = 17 and 87, respectively), indicating EMT and metastasis-promoting the role of miR-199a-5p. In contrast, let-7a-5p expression was significantly downregulated in the higher (8–9) compared to lower (6–7) Gleason score cohort (N = 17 and 87, respectively), indicating its potential anti-metastatic role in PCa. (**F**,**G**) Comparison between the two cohorts showing that miR-199a-5p expression is significantly upregulated in the higher (8–9) compared to lower Gleason score (6–7) samples (N = 17 and 87, respectively), indicating the EMT and metastasis-promoting the role of miR-199a-5p. In contrast, let-7a-5p expression was significantly downregulated in the higher (8–9) compared to lower (6–7) Gleason score cohort (N = 17 and 87, respectively), indicating its potential anti-metastatic role in PCa. * (*p* < 0.05), # (*p* < 0.01). Data are presented as the mean ± SD.

**Figure 8 cancers-14-01625-f008:**
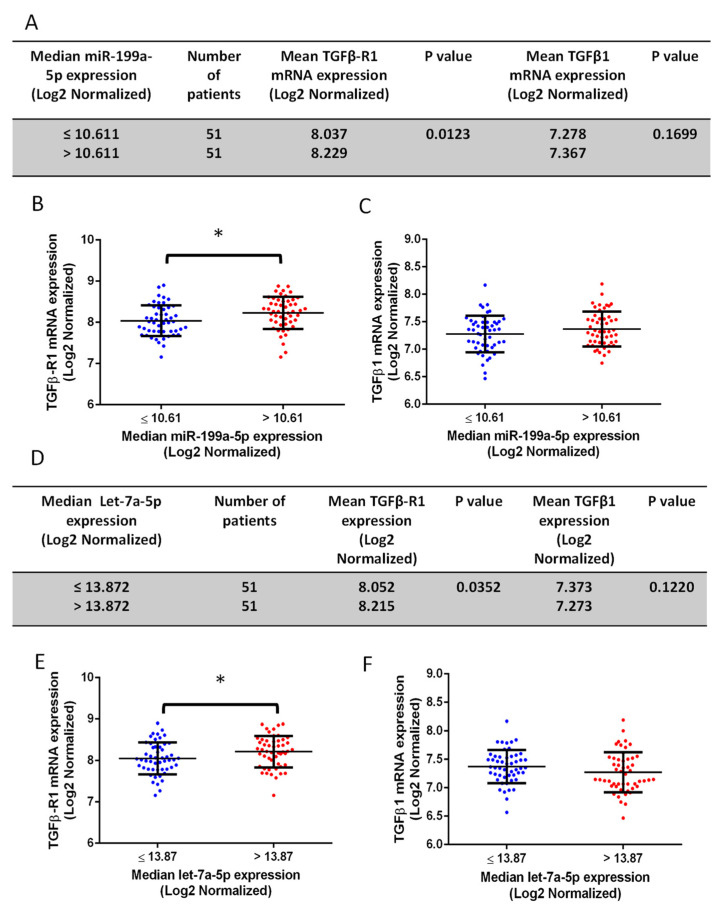
High miR-199a-5p and low let-7a-5p expression correlate with increased TGFβ1 and TGFβ-R1 expression. (**A**) Data showing the mean TGFβ-R1 and TGFβ1 expression from the TCGA study based on the median miR-199a-5p expression. (**B**,**C**) Comparison based on the median miR-199a-5p expression of 102 patients out of the 113 total patients (11 were excluded; 7 with unknown Gleason score, 2 outliers and 2 with unknown TGFβ-R1 and TGFβ1 expression), showing a correlation between higher miR-199a-5p expression and higher TGFβ-R1 expression associated with a positive trend towards increased TGFβ1 (Log2 normalized) expression. (**D**) Data showing the mean TGFβ-R1 and TGFβ1 expressions from the TCGA study based on the median let-7a-5p expression. (**E**,**F**) A comparison between high and low let-7a-5p expression based on the median value for 102 patients showed lower let-7a-5p expression significantly associated with lower TGFβ-R1 expression and a positive trend towards increased TGFβ1 expression. * (*p* < 0.05). Data are presented as the mean ± SD.

**Figure 9 cancers-14-01625-f009:**
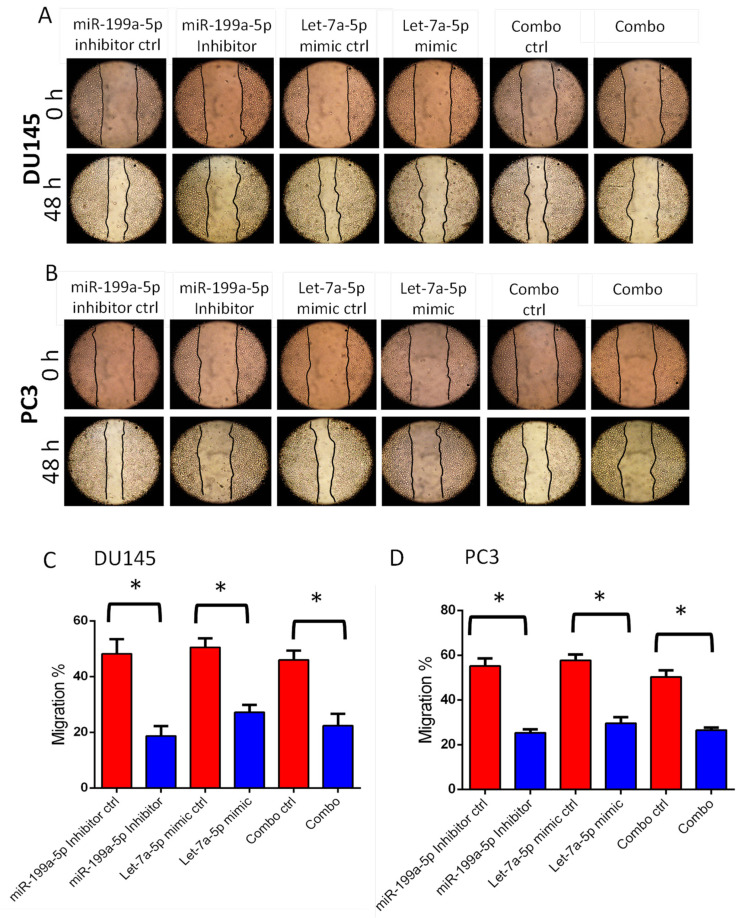
miR-199a-5p inhibition and/or let-7a-5p activation impairs DU145 and PC3 cell motility. (**A**,**B**) Representative images from the scratch assay (0 and 48 h) for DU145 and PC3 cells show the regulatory role of miR-199a-5p and let-7a-5p on cell migration (*n* = 3) (**C**,**D**). 25 nM of miR-199a-5p inhibitor and/or 10nM of let-7a-5p mimic significantly inhibited the motility of both DU145 and PC3 cells (*n* = 3)]. * *p* < 0.001. Data are presented as the mean ± SD. Matched miRNAs were further analyzed for their potential targets using DIANA-mirPath v 3.0 and dbEMT databases (Figure 3A–D) for their potential regulating of EMT gene expression as presented in volcano plots (Figure 3A,B), and compared to the NanoString^®^ mRNA analysis (Figure 4 and Figure 5).

**Figure 10 cancers-14-01625-f010:**
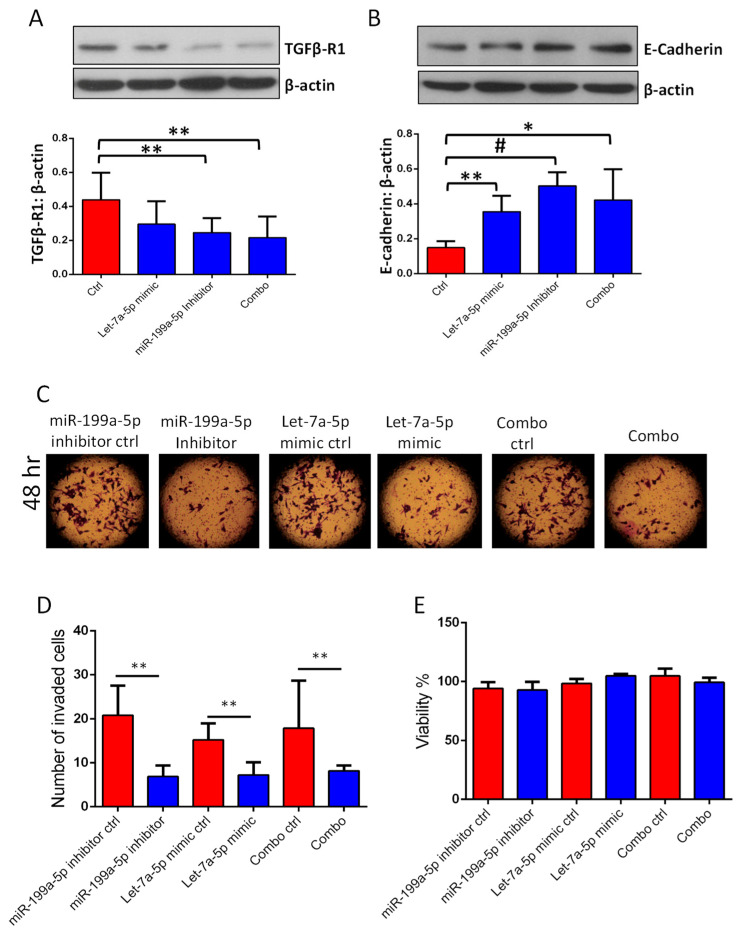
Inhibition of miR-199a-5p and activation of let-7a-5p inhibited TGFβ-R1 expression, EMT, and DU145 cell invasion. (**A**,**B**) Representative Western blot images and bar graph showing the band densitometry analysis of miR-199a-5p inhibitor (25nM) and/or let-7a-5p mimic (10 nM) effects in DU145 cells on TGFβ-R1 (ALK5) and E-Cadherin expression (*n* = 3). (**C**) The use of miR-199a-5p inhibitor (25 nM) and/or let-7a-5p mimic (10nM) in DU145 cells enhanced E-Cadherin expression (*n* = 3). (**D**) miR-199a-5p inhibition and/or let-7a-5p activation in DU145 cells resulted in impaired cell migration (*n* = 3). (**E**) Neither miR-199a-5p inhibition and/or let-7a-5p activation in DU145 cells had any significant effect on cell viability (*n* = 3). * (*p* < 0.05); ** (*p* < 0.01); # (*p* < 0.001); unpaired Student t-test for two groups analysis (GraphPad Prism 6.01). Data are presented as the mean ± SD.

## Data Availability

All the data are included in the manuscript.

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
