# Peer review of "Regulation of Let-7a-5p and miR-199a-5p Expression by Akt1 Modulates Prostate Cancer Epithelial-to-Mesenchymal Transition via the Transforming Growth Factor-β Pathway"

_cancers, 2022, doi:10.3390/cancers14071625_

Round 1
Reviewer 1 Report
This interesting and well-written manuscript describes the regulation of Let-7a-5p and miR-199a-5p expression by Akt1 and its effects on TGFbeta and EMT in prostate cancer. I have a few issues that need to be addressed:
- The choice of the AR-negative PC3 and DU145 PCa cells is not clearly justified. While these are metastatic cell lines (bone and brain respectively), they may not correlate clinically with the patient transcriptomic data analyzed from the cBioportal, which I presume was derived from non-metastatic, radical prostatectomy derived prostate tumors. It is not clear why earlier stage, AR+ metastatic cell lines (e.g., LnCaP, VCAP, 22RV1, MDA-PC-2b) were not included in the analyses.
- Likewise, the choice of the MSKCC cBioportal dataset was not clearly justified given the numerous PCa datasets in this portal. Why this one in particular? Did the authors find in cBioportal other PCa datasets that have tissues from metastatic lesions?
- Did the nanostring analyses reveal other significantly regulated pathways related to EMT but independent of TGFbeta?
- The difference between the two nanostring experiments (Fig. 1 and Fig. 4) does not come across clearly. Why two different but almost identical nanostring analyses?
- Figure 7 is very confusing and difficult to interpret given the close proximity of the panels to each other, the small font size, and the combination of figures and tables. These should be separated more with larger font sizes, and discussed more clearly in the legend. For instance, panel H is not mentioned in the legend. It doesn't come across clearly how panels E-H correspond to the different Gleason cohorts.
- While the effects of miR-199 inhibition and let-7 activation in PC3 and DU145 cells were assessed by invasion and migration assays, there was a lack of other assays associated with aggressive properties of these cells such as colony formation and tumorsphere formation.
- Through the first 7 Figures, TGFBR1 is identified, however, in Figures 9 and 10 Western blots for this protein are labeled as ALK5. This is inconsistent and could be confusing.
- Figures 9 and 10 are identical in both panels and results, except that one corresponds to DU145 cells and the other to PC3 cells. One of these figures should be cited in the text but included in the Supplementary materials.
Author Response
- The choice of the AR-negative PC3 and DU145 PCa cells is not clearly justified. While these are metastatic cell lines (bone and brain respectively), they may not correlate clinically with the patient transcriptomic data analyzed from the cBioportal, which I presume was derived from non-metastatic, radical prostatectomy derived prostate tumors. It is not clear why earlier stage, AR+ metastatic cell lines (e.g., LnCaP, VCAP, 22RV1, MDA-PC-2b) were not included in the analyses.
Response: This is a very good point. We have justified in the revised manuscript. Our work has been focused on the AKT1-regulated miRNAs linking to the TGFβ signaling and regulation of EMT. Hence, we specifically looked for genes relating to EMT and metastasis. We did not pick LNCaP cells as our previous studies demonstrated LNCaP cells did not undergo EMT in response to TGFβ treatment (PMID: 25714023). We agree that Akt1 is involved in regulating AR expression as shown by Nissim Hay’s group in (PMID 16778075). However, the paper reported that, besides nuclear localization of AR in the murine prostate cancer tissues, there was a decrease in AR expression in Akt1 knockout mice, indicating that the involvement of AR signaling would be minimal. Hence, we chose AR-negative PC3 and DU145 PCa cells.
- Likewise, the choice of the MSKCC cBioportal dataset was not clearly justified given the numerous PCa datasets in this portal. Why this one in particular? Did the authors find in cBioportal other PCa datasets that have tissues from metastatic lesions?
Response: Once again, this is a great question, and we have justified it here as well. The reason we picked the MSKCC 2010 study is that it is the only source of miRNA datasets on prostate cancer in cBioportal. In the past, we have used several cBioportal database studies for bioinformatics analysis of expression of Akt isoforms and TGFbeta isoforms in cancer, which are published (PMID: 31360736).
- Did the nanostring analyses reveal other significantly regulated pathways related to EMT but independent of TGFbeta?
Response: We agree with the reviewer that Nanostring analysis revealed several microRNAs and mRNAs belonging to various pathways. Investigating every pathway can be overwhelming for a focused project. Hence, we analyzed the data from NanoString using the Ingenuity Pathway Analysis (IPA) software and other databases to screen and identify the miRNAs and mRNAs specifically involved in regulating the TGFβ pathway as well as their impact on cancer invasiveness and progression. All genes that significantly changed with AKT1 inhibition and were revealed by Nanostring were involved in the analysis. We agree that other pathways have also modulated with Akt1 suppression in PCa, however, is outside of the scope of the current paper. We have included this as a limitation of our study in the ‘conclusions’ section.
- The difference between the two nanostring experiments (Fig. 1 and Fig. 4) does not come across clearly. Why two different but almost identical nanostring analyses?
Response: The scheme of Nanostring analysis in Figures 1 and 4 are different and are two separate assays and analyses. In figure 1, the scheme presents how we analyzed and narrowed down to our investigation on miRNAs, and in figure 4, the scheme presents the analysis we did for mRNAs. Thus, in this paper, we present a unique study comparing miRNA and mRNA expressions from the same cells, followed by bioinformatics and database analysis of the Akt1-miRNA-TGFbeta-EMT pathway in prostate cancer invasion.
- Figure 7 is very confusing and difficult to interpret given the close proximity of the panels to each other, the small font size, and the combination of figures and tables. These should be separated more with larger font sizes, and discussed more clearly in the legend. For instance, panel H is not mentioned in the legend. It doesn't come across clearly how panels E-H correspond to the different Gleason cohorts.
Response: We apologize. In addition, we realized that some information from the figure legend was lost while transferring from the information to the journal template and missed explaining panels E, F, G, and H. To provide more clarity to each panel, we have split this figure into two (Figures 7 and 8 in the revised manuscript) and have included additional information to each figure legend.
- While the effects of miR-199 inhibition and let-7 activation in PC3 and DU145 cells were assessed by invasion and migration assays, there was a lack of other assays associated with aggressive properties of these cells such as colony formation and tumor-sphere formation.
Response: As mentioned above, the objective of this study is to assess the changes in the migratory and invasive ability of prostate cancer cells (because of EMT) by miRNAs that are found to be regulated by Akt1 in our study. Since EMT is associated with increased motility and invasiveness of cells, we focused on these two assays. We agree with the reviewer that changes in other tumor cell functions because of modulations in other pathways cannot be ruled out. This, however, need additional studies, which is outside the scope of the current study. We have included this in the study limitations in the ‘conclusions’ section.
- Through the first 7 Figures, TGFBR1 is identified, however, in Figures 9 and 10 Western blots for this protein are labeled as ALK5. This is inconsistent and could be confusing.
Response: We apologize for this confusion. We have replaced ALK5 with TGFBR1 in the figures legends and the text.
- Figures 9 and 10 are identical in both panels and results, except that one corresponds to DU145 cells and the other to PC3 cells. One of these figures should be cited in the text but included in the Supplementary Materials.
Response: We have moved the Figure on PC3 cells to the Supplementary materials as suggested.
Reviewer 2 Report
In this study, the authors explore molecular mechanisms underlying metastasis in the context of prostate cancer with a focus on TGFbeta signaling and its functional relationship with Akt1. In particular, the authors focus on the observation that suppression of Akt1 promotes invasive/metastatic behavior of prostate cancer cells, and explore microRNAs that may mediate the consequences of Akt1 suppression. The goal of the study is to address gaps in knowledge of how PCa metastasizes from the primary tumor, and to ultimately use this new information for therapeutic benefit. The authors used Nanostring-based profiling to assess miRNAs and mRNAs that were differentially altered following shRNA-based silencing of Akt1 in PC3 or DU145 cell lines, and used mirPath, dbEMT and cBioPortal to identify two miRNAs – let-7a-5p and miR-199a-5p – that were linked to TGFb signaling and EMT. The authors proceeded to evaluate the level of let-7a-5p and miR-199a-5p in experimental lung metastases of DU145 cells in mice and in human PCa tissues of different Gleason score and suggest that miR-199a-5p is increased and let-7a-5p is decreased in Gleason high vs Gleason low specimens. Next, the impact of let-7a-5p mimetic and miR-199a-5p inhibitor on cell migration and invasion in vitro was assessed, with both miRNA manipulations shown to reduce migration. Specific comments are noted below.
- Since the authors are interested in the molecular mechanisms that lead to acquisition of EMT and promotion of metastasis, it would have been interested to profile cell lines that were not derived from metastasis, but rather reflected a pre-metastatic and post-metastatic state e.g. Can the authors better justify the choice of PC3 and DU145 cell lines for the discovery experiments?
- In relation to the data in Figure 7, what was the Akt1 expression status in the prostate tumor specimens, in relation to the let-7a-5p, miR-199a-5p and TGFbeta receptor and ligand?
- In Figure 7A (middle panel), the qPCR data for low and high Gleason score prostate tumors are presented as ‘Fold change’ but it is unclear what the reference group is? Fold change relative to what? It is also unclear why the middle panel is presented as a bar graph and the right panel presented as a scatter plot. Please clarify.
- In Figure 7E, median expression of let-7a-5p and miR-199a-5p compared to mean expression in Figure 7B/C whereas mean TGFb1R/TGFb1 mRNA is compared. It is unclear why the different comparisons are using different summary statistics. Lastly, although the differences are calculated to be statistically significant, are the differences in TGFb1 mRNA biologically meaningful?
- In relation to Figures 8 -10, what is the consequence of miR199 mimic or let-7a-5p inhibitor on migration and invasion? In other words, in cells in which Akt1 is intact, does forced expression of miR99 mimic or inhibition of let-7a-5p lead to a similar phenotype to shAkt1 in PC3 and DU145? These are important controls to understand the functional relevance of miR199a and let-7a-5p.
- It would be helpful if the authors could include a working model to summarize the impact of Akt1 silencing, induction of miRNAs, attenuation of mRNAs and the manifestation of EMT/migration and invasion phenotypes.
Author Response
- Since the authors are interested in the molecular mechanisms that lead to acquisition of EMT and promotion of metastasis, it would have been interested to profile cell lines that were not derived from metastasis, but rather reflected a pre-metastatic and post-metastatic state e.g. Can the authors better justify the choice of PC3 and DU145 cell lines for the discovery experiments?
Response: Great question. As responded to reviewer #1, our study is focused on the AKT1-regulated miRNAs and their crosstalk with the TGFβ signaling in the regulation of EMT. Hence, we specifically focused on genes relating to EMT and metastasis. Based on our previous experience, early-stage cancers cells such as the LNCaP cells did not undergo EMT in response to TGFβ treatment (PMID: 25714023). For this reason, we resorted to metastatic cancer cell lines for molecular characterization. We have justified in the revised manuscript.
- In relation to the data in Figure 7, what was the Akt1 expression status in the prostate tumor specimens, in relation to the let-7a-5p, miR-199a-5p and TGFbeta receptor and ligand?
Response: Great question. This was done but we found no difference at the mRNA levels of Akt1. Please see this new data included in the supplemental figures. This is likely because Akt is regulated primarily by phosphorylation rather than expression. We have previously demonstrated this through a similar cBioportal based study (PMID: 31360736).
- In Figure 7A (middle panel), the qPCR data for low and high Gleason score prostate tumors are presented as ‘Fold change’ but it is unclear what the reference group is? Fold-change relative to what? It is also unclear why the middle panel is presented as a bar graph and the right panel presented as a scatter plot. Please clarify.
Response: All fold changes are relative to SNORD61 expression in the same set of samples in each group/column. The information has been included in the figure 7 legend. As suggested by the reviewer, we have also replaced the middle panel with a scatter plot matching to the right panel.
- In Figure 7E, median expression of let-7a-5p and miR-199a-5p compared to mean expression in Figure 7B/C whereas mean TGFb1R/TGFb1 mRNA is compared. It is unclear why the different comparisons are using different summary statistics. Lastly, although the differences are calculated to be statistically significant, are the differences in TGFb1 mRNA biologically meaningful?
Response: in panels 7 E-H, it would be more accurate to use the median of miRNA to split into 2 groups; those with high expression and those with low expression, and compare TGFB and TGFBR1 on that basis. While in panels B-D, the independent variables are Gleason scores (high or low), so using the mean or median expression of the miRNAs would not make a difference. Regarding the biological relevance of the clinical data, this is a usual case with such data mining studies. While the data provides a trend, the confirmations come experimentally.
- In relation to Figures 8 -10, what is the consequence of miR199 mimic or let-7a-5p inhibitor on migration and invasion? In other words, in cells in which Akt1 is intact, does forced expression of miR99 mimic or inhibition of let-7a-5p lead to a similar phenotype to shAkt1 in PC3 and DU145? These are important controls to understand the functional relevance of miR199a and let-7a-5p.
Response: In Akt1 intact PCa cells, we expect overexpression of miR-199a-5p and inhibition of let-7a-5p expression to result in a similar phenotype of ShAkt1 PC3 or DU145 cells. Although including these experiments would be good, the time, effort, and cost would also be demanding. Nevertheless, the study arms we have included, and the confirmation of signature gene and microRNA expression changes in two different cell lines, clearly demonstrate the role of the Akt1-miRNA-TGFbeta-EMT pathway in prostate cancer motility and invasion.
- It would be helpful if the authors could include a working model to summarize the impact of Akt1 silencing, induction of miRNAs, attenuation of mRNAs and the manifestation of EMT/migration and invasion phenotypes.
Response: This is a great suggestion. We have now included a separate figure explaining the overall conclusions from the study. Please see Supplemental Figure 7.
Reviewer 3 Report
There is no doubt that current manuscript is novel in terms of title and it will make great advances in field of cancer therapy and basic molecular research that will be of importance for clinical courses in near future. It has focused on miRNAs as a key member of ncRNAs and prostate cancer as a malignant urological cancer. Furthermore, current article focuses on EMT as a factor involved in metastasis in prostate cancer. Overall, current manuscript has high quality and I recommend its publication. However, issues should be addressed before publication. The methods and results are ok and reliable. In figure 1, authors have provided a schematic figure. There are two options: a) authors can keep the figure and add a high quality figure of prostate cancer and related molecular pathways or b) authors can replace figure 1 with a high quality one. What are the limitations of current work and they should be mentioned in conclusion. What are clinical application of current work and they should be mentioned in conclusion section. The first paragraph of introduction is about prostate cancer. However, more specific information about prostate cancer, its malignancy and so on should be provided. Only one reference from 2020 and one reference from 2021. Then, update references. Add color to some diagrams to enhance their quality. Remove second paragraph of introduction. Provide specific and updated information about TGF-beta and EMT and associate them with each other. Current representation makes confusion in readers. The understanding of section related to miRNAs is easy for a researcher working in field of non-coding RNAs, not a general reader. Please add more basic information about mIRNAs, their location, their involvement in regulation of biological mechanism, their regulation by upstream mediators and so on. Suggested article could be Doi, 10.1016/j.canlet.2021.03.025. Add more statements in discussion to elaborate your work.
Author Response
- In figure 1, authors have provided a schematic figure. There are two options: a) authors can keep the figure and add a high quality figure of prostate cancer and related molecular pathways or b) authors can replace figure 1 with a high quality one.
Response: The schematic provided in figure 1 explains the flow of analysis that we performed starting from the NanoString analysis of miRNAs, their statistical analysis for significant changes, identification of common miRNAs modulated by Akt1 loss in both cell types, their characterization using bioinformatics analysis, and picking the candidate miRNAs for the characterization of their cellular function and clinical relevance. Since the figure is already crowded, we believe that adding another panel to this may confuse the readers. The authors feel that an additional panel with molecular pathways regulating prostate cancer would not fit here at the beginning of a research article. Instead, we have generated a working hypothesis figure and have been included it in the supplemental figures.
- What are the limitations of current work and they should be mentioned in conclusion. What are clinical application of current work and they should be mentioned in conclusion section.
Response: We have revised the conclusions section to incorporate the above-suggested changes.
- The first paragraph of introduction is about prostate cancer. However, more specific information about prostate cancer, its malignancy and so on should be provided.
Response: We have revised the 1st paragraph in the introduction section.
- Only one reference from 2020 and one reference from 2021. Then, update references.
Response: We have updated the bibliography with more recent articles.
- Add color to some diagrams to enhance their quality.
Response: Thanks for this suggestion. We have added colors to 4 figures in the revised manuscript.
- Remove second paragraph of introduction. Provide specific and updated information about TGF-beta and EMT and associate them with each other. Current representation makes confusion in readers.
Response: We have revised the 2nd paragraph of the introduction section incorporating the reviewer's suggested changes.
- The understanding of section related to miRNAs is easy for a researcher working in field of non-coding RNAs, not a general reader. Please add more basic information about mIRNAs, their location, their involvement in regulation of biological mechanism, their regulation by upstream mediators and so on.
Response: We have revised the introduction section to include the reviewer suggested information.
Round 2
Reviewer 2 Report
No further comments.